

# Applying machine learning methods to prediction problems of lattice observables

**N.V. Gerasimeniuk[1,*], M.N. Chernodub[1,2], V.A. Goy[1,2], D.L. Boyda[3], S.D. Liubimov[1] and A.V. Molochkov[1]**

**1** Pacific Quantum Center, Far Eastern Federal University, 690950, Vladivostok, Russia
**2** Institute Denis Poisson CNRS/UMR 7013, Universite' de Tours, 37200 France
**3** Argonne Leadership Computing Facility, Argonne National Laboratory,
Lemont IL-60439, USA

* gerasimenyuk_nv@dvfu.ru

*Proceedings for the XXXIII International Workshop on High Energy Physics,
Hard Problems of Hadron Physics: Non-Perturbative QCD & Related Quests
ONLINE, 8-12 November 2021*

## Abstract

**We discuss the prediction of critical behavior of lattice observables in SU(2) and SU(3) gauge theories. We show that feed-forward neural network, trained on the lattice configurations of gauge fields as input data, finds correlations with the target observable, which is also true in the critical region where the neural network has not been trained. We have verified that the neural network constructs a gauge-invariant function and this property does not change over the entire range of the parameter space.**

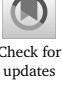
## 1 Introduction

The low-energy physics of strong interactions cannot be addressed analytically because of the strong coupling, which makes perturbative approaches, usually used in the high-energy region, unreliable. For that reason, all existing calculations in the non-perturbative domain are based on effective low-energy models or sophisticated numerical methods involving the Monte Carlo (MC) algorithms. The MC simulations are reasonably reliable to address various thermodynamic properties of quantum chromodynamics (QCD) from the first principles. In the physically relevant domain of parameters, the numerical simulations are very computationally expensive and thus require powerful supercomputers.

In addition, the Monte Carlo methods cannot be applied to the interesting region of the QCD phase diagram at finite baryon chemical potential, thus calling for the development of alternative approaches aimed, in particular, at the investigation of the quark-gluon plasma at

finite density. A promising way to extend the MC techniques involves the machine learning (ML) methods used nowadays to address various problems in physics [1, 2].

Our work discusses an example of the potentially helpful combination of the machine learning techniques with the standard MC methods. In the context of lattice field theory, the synergy of these two approaches remains largely unexplored. The use of machine learning methods is mainly reduced to (i) the investigation of the ability of neural networks to predict lattice observables in non-perturbative domains of parameters and (ii) generate lattice field configurations as an alternative to the generally accepted Monte Carlo approach. In our paper, we address the question of the critical behavior of lattice observables in SU(2) and SU(3) gauge theories with the ML techniques that respect the gauge-invariant structure of the theory.

## 2 Machine learning

The use of machine learning methods in lattice QCD is reduced to solving several problems: regression problem, classification problem and simulation problem.

Simulations of configurations in lattice QCD are often computationally expensive, that complicates the process of accumulating statistical data. Modern machine learning techniques can provide opportunities to improve simulation speed. One can build neural network to simulate lattice configurations and after training this approach require less computer power and time than common methods to simulate lattice configurations [3, 4].

In case of searching for new physics we try to solve regression or classification problem where neural network trains to reconstruct certain observables from a given information of Monte Carlo configurations corresponding to some set of lattice parameters. A well-trained neural network is subsequently able to predict the value of the observable from data that was previously unknown to it [5, 6]

But, generally accepted ML methods aimed to solving various problems in the field of computer vision are not suitable for solving problems in the field of gluodynamics, since lattice data is another kind of data that is fundamentally different from classical images. In order to use machine learning in LQCD, it is necessary either to take into account the properties of lattice data within the neural networks itself or transform the lattice data to a more convenient form. The construction of neural networks consistent with the local symmetry and the matrix origin of the lattice data is still an active topic of discussion in the current literature [7, 8].

## 3 Lattice Simulations

In this work, we carry out simulations of non-Abelian Yang-Mills gauge theory in lattice regularization with two and three colors. We use Wilson discretization of action

$$S[U] = \beta \sum_P \left( 1 - \frac{1}{N} \text{Re} \left[ \text{Tr} \, U_P \right] \right), \tag{1}$$

where $\beta$ is theory parameter that correspond to lattice gauge coupling, $N$ defined number of colors and $U_P = U_{x,\mu} U_{x+\hat{\mu},\nu} U^{\dagger}_{x+\hat{\nu},\mu} U^{\dagger}_{x,\nu}$ is plaquette variable which build from original link variables $U_{x,\mu} \in SU(N)$. Wilson action is formulated in the Euclidean spacetime on the lattice with the volume $N_s^3 \times N_t$. We are used periodic boundary conditions in all directions. For study dependence from lattice volume we use $N_t = 2, 4$ and $N_s = 8, 16, 32$. The partition function of our system is defined with the formula

$$Z = \int dU \, e^{-S[U]}. \tag{2}$$

There are two phases of this theory: confinement and deconfinement. Confinement corresponds to small values of the coupling constant $\beta$, deconfinement to high values. In the case of two-color gluodynamics, these two phases are separated by a second-order phase transition. In the case of $N \geq 3$, a first-order phase transition is observed.

The well-known order parameter of the deconfinement phase transition is the Polyakov loop. In the lattice calculations, it is convenient to identify the bulk Polyakov loop:

$$L = \frac{1}{N_s^3 N} \Big\langle \sum_x \text{Tr} \prod_{t=0}^{N_t - 1} U_{x,t;4} \Big\rangle, \tag{3}$$

where the sum goes over all spatial sites $x$ of the lattice.

## 4 Neural network architecture and training process

We are trying to find such an architecture of a neural network that could catch correlations with a targeted observable (Polykov loop) and display its properties. In this section, we describe the machine-learning algorithm which includes building of the architecture and training of the neural network. The training points for SU(2) and SU(3) gauge theories are set at the lattice coupling constants $\beta = \beta_{SU(2)} = 4$ and $\beta = \beta_{SU(3)} = 10$, respectively. Both these points correspond to a deep weak-coupling regime.

The values $\beta_{SU(2)}$ and $\beta_{SU(3)}$ were chosen by us because explicit calculations of the Polyakov loop are still possible at these points with the support of Monte Carlo calculations. In another turn, these points correspond to the region where the Polyakov loop has a nonzero behavior, which is important for a qualitative training process. It is also well known that in this region the Polyakov loop has several vacuum states, depending on the theory. As we will show, based on this information the neural network does not need more knowledge to reconstruct the behavior of the predicted order parameter.

Referring to the problem described above, in order to build a neural network architecture, we need to transform the input multidimensional lattice data in the most convenient way. In this project we propose to use the vector representation of matrices, however, in the case of $SU(2)$ gauge fields, we use only 4 components, since the rest of the matrix components are highly correlated. The vector representation is the following for $SU(2)$ matrix:

$$U = \begin{pmatrix} u_{11} & u_{12} \\ u_{21} & u_{22} \end{pmatrix} \equiv \begin{pmatrix} a_1 + i a_2 & a_3 + i a_4 \\ -a_3 + i a_4 & a_1 - i a_2 \end{pmatrix} \rightarrow \begin{pmatrix} a_1 \\ a_2 \\ a_3 \\ a_4 \end{pmatrix}, \tag{4}$$

where $a_1 = \text{Re}(u_{11})$, $a_2 = \text{Im}(u_{11})$, $a_3 = \text{Re}(u_{12})$, and $a_4 = \text{Im}(u_{12})$. In the case of the $SU(3)$ theory, the full set of matrix values is used.

After preparing the input data, we have an input lattice tensor with the following shape $(N_t, N_s, N_s, N_s, Dim, U)$. The last dimension of this tensor is the elements of the matrix in the corresponding vector form. $Dim$ is the index of $\mu$-direction for the matrix $U_\mu(x)$, the indices $N_s$ and $N_t$ represent the number of sites in the lattice configuration for the spatial and temporal direction. Also worth noting that we use 3D convolution layers and for that reason we reshape the input lattice data to 4D where the last dimension correspond to the channels of neural network. The first two spatial directions are merged because element $U[x][y]$ can be represented as $U[y * N_s + x]$ by cost of locality. The last two dimensions can also merged into one, since we are interested in correlations between matrices located in neighboring links. As result the architecture of the neural network are the following sequence of layers: convolution,

relu activation, average pooling, flatten and dense layer. Using different sizes of lattice data requires new architectures to be built, as it turned out, an increase in the temporal direction in the input data requires an additional convolution-relu sequence. The final architecture of neural networks are presented in the Table 1.

Table 1: Neural network architectures for various size of input configurations.

| InputData($N_t = 2, N_s^2, N_s, U_\mu$) | InputData($N_t = 4, N_s^2, N_s, U_\mu$) |
| :---: | :---: |
| Conv3D + ReLU | Conv3D + ReLU + Conv3D + ReLU |
| AveragePooling3D + Flatten | AveragePooling3D + Flatten |
| Dense | Dense |

Before starting the training process, we generate 9000 configurations with different parameters $N_s$ and $N_t$ at one fixed coupling constant $\beta_{SU(2)}$ or $\beta_{SU(3)}$. During the training process, we also guarantee that the data from each vacuum state will be used in the same proportion. For prediction the Polykov loop we generate other 100 configurations at points with $\beta \leq \beta_{SU(2)}$ or $\beta \leq \beta_{SU(3)}$. In SU(3) case the Polyakov loop becomes complex number, here we predict the real and imaginary parts separately. We choose the mean squared error (MSE) as a loss function. The neural network parameters' optimization algorithm is Adam-method.

As a result, we built and trained a neural network that can predict the behavior of the Polyakov loop in the $\beta < \beta_{SU(N)}$ region based on the knowledge from only one $\beta$ point. In Figure 1 we demonstrate the effectiveness of a neural network algorithm for qualitative prediction of the order parameter over the entire range of the coupling constant.

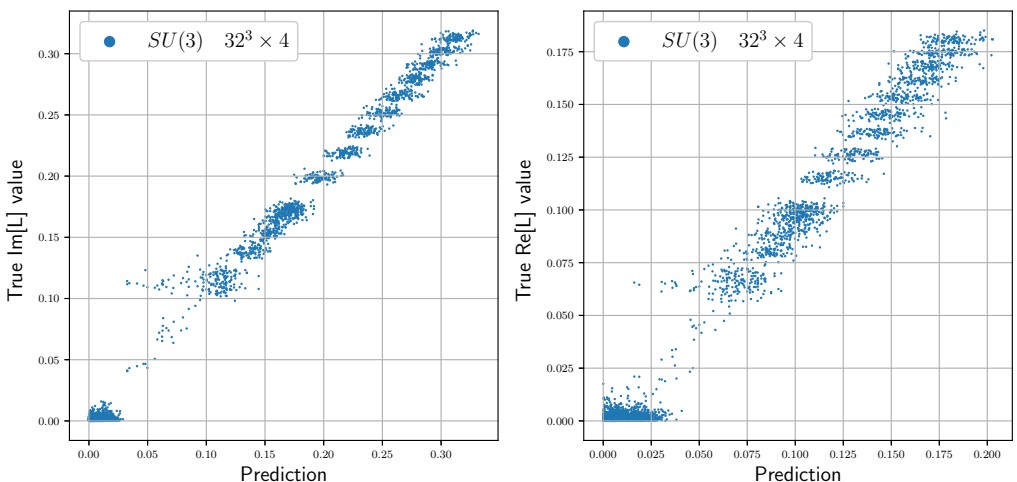

Figure 1: Prediction of neural network on a sample by sample basis of SU(3) $32^3 \times 4$ configurations, where $\beta$ values were used in the range from 4 to 7 with a step of 0.2.

Another important aspect of this prediction is the verification of the invariance of a given observable with respect to gauge transformations. For this check, we completely change the configuration using a set of different uniformly distributed SU(2) or SU(3) matrices respectively. We do several changes and make a prediction for each step for the already changed configuration. In Figure 2, we demonstrate the result for such a test for the case of the SU(2) theory and lattice size equal to $N_t = 4, N_s = 16$.

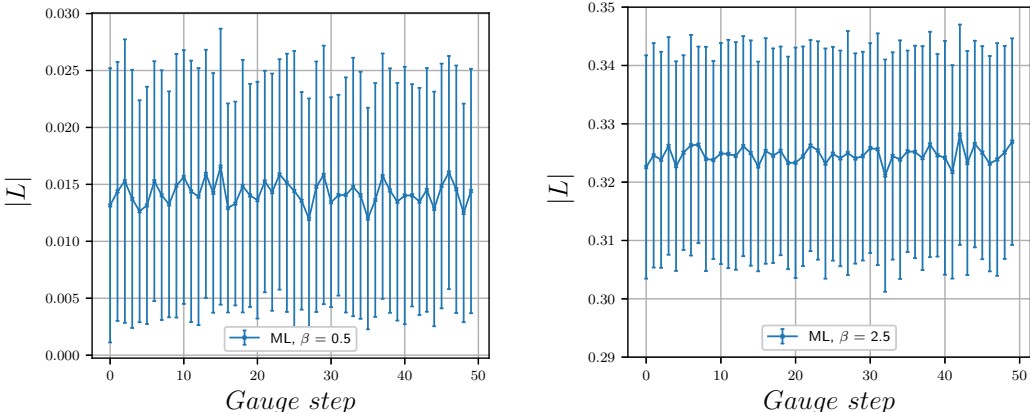

Figure 2: Gauge invariant behavior of numerically constructed Polykov loop at different phases of SU(2) theory as function of uniformly distributed global random gauge transformation step.

## 5   Conclusion

We demonstrate that the machine-learning algorithms allow us to restore, using the data from an unphysical point of the lattice parameter space, the gauge-invariant order parameter applicable to the whole physical critical region of the theory. In other words, our neural network is able to the physical order parameter relevant to the numerically costly critical regime of the model after a training procedure at a set of lattice field configurations that were generated by fast Monte Carlo methods at a single unphysical point outside of the continuum limit of the lattice model.

We also demonstrated that the classical feed-forward neural network could be used to restore simple observables and predict their properties in the critical region of the theory. Our work potentially implies that the ML techniques can predict other, more complex observables and thus be applied to the regions which are unreachable to the standard MC methods. In addition, we demonstrated how the non-gauge-invariant architecture of the deep learning model may produce gauge-invariants solutions within statistical uncertainties.

## Acknowledgements

The work of M.N.C, N.V.G, V.A.G, S.D.L, and A.V.M was supported by the grant of the Russian Foundation for Basic Research No. 18-02-40121 mega and partially carried out within the state assignment of the Ministry of Science and Higher Education of Russia (Project No. 0657-2020-0015). The numerical simulations of Monte Carlo data were performed at the computing cluster Vostok-1 of Far Eastern Federal University. Scientific calculations and analysis by Denis Boyda were supported by the Argonne Leadership Computing Facility, which is a U.S. Department of Energy Office of Science User Facility operated under contract DE-AC02-06CH11357.

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
