# Peer review of "Applying machine learning methods to prediction problems of lattice observables"

_SciPost Physics Proceedings, doi:SciPost Phys. Proc. 6, 020 (2022)_

## Round 2 · Referee Report · Anonymous (Referee 1) · 2022-3-17

Strengths

A combination of machine learning techniques and the standard Monte Carlo simulations is proposed as a method to solve hard problems of the lattice QCD
The original approach to the neural network formualtion is proposed.
The numerical results obtained clearly demonstrate the strong potential of the proposed method.

Weaknesses

The misprints and English Grammer should be corrected

Report

The paper is devoted to the study of the lattice QCD using a combination of machine learning techniques and the standard Monte Carlo simulations. The application of machine learning to the lattice QCD is quite new and promising. The motivation for this work is that the Monte Carlo technique is expensive and sometimes (at nonzero quark chemical potential) its application is extremely costly.

The authors study the SU(2) and SU(3) gluodynamics at finite temperature including the vicinity of the respective finite temperature phase transitions.
The lattice field configurations generated by the Monte Carlo method at unphysically large values of $\beta$ are used as input for the neural network. The values of the Polyakov loop at values of $\beta$ in the physically interesting range are then computed with the help of the neural network. This result demonstrates the potential of the combination of the Monte Carlo technique and the ML technique in solving the hardest problems of the lattice QCD simulations like strong autocorrelations in the critical region or the so-called sign problem at nonzero baryon chemical potential.
In my opinion the paper presents new very promising approach to the lattice QCD simulations and definitely deserves to be published after minor changes.

Requested changes

It is necessary to provide the values of $\beta$ used in Fig.1

Here is the list of expressions which should be corrected

p.2
which build from original link variables
N defined number of colors
For study dependence from lattice volume
We are used periodic boundary conditions

p.3
has a non-zero behavior
The last dimension of this tensor is the elements
Also worth noting
The last two dimensions can also merged into one

p.4
located in neighboring links
the architecture of the neural network are the following sequence of layers
For prediction the Polykov loop

p.5
our neural network is able to the physical order parameter
gauge-invariants solutions

  • validity: good
  • significance: high
  • originality: high
  • clarity: good
  • formatting: reasonable
  • grammar: below threshold

Author:  Nikolai Gerasimeniuk  on 2022-03-31  [id 2342]

(in reply to Report 1 on 2022-03-17)
Category:
answer to question

Our response: 1. It is necessary to provide the values of $\beta$ used in Fig.1: In Fig.1 $\beta$ values are in the range from 4 to 7 with a step of 0.2. The relevant changes will be added in the resubmission. 2. Here is the list of expressions which should be corrected... We agree with all your comments and are ready to correct them in the resubmission.

---

## Editorial Decision

published